# Impact of the COVID-19 Pandemic on the Incidence and Thickness of Cutaneous Melanoma in Belgium

**DOI:** 10.3390/biomedicines11061645

**Published:** 2023-06-06

**Authors:** Philip Georg Demaerel, Arthur Leloup, Lieve Brochez, Liesbet Van Eycken, Marjan Garmyn

**Affiliations:** 1Department of Dermatology, University Hospitals Leuven, 3000 Leuven, Belgium; marjan.garmyn@uzleuven.be; 2Belgian Cancer Registry, 1210 Brussels, Belgium; arthur.leloup@kankerregister.org (A.L.); elizabeth.vaneycken@kankerregister.org (L.V.E.); 3Department of Dermatology, Ghent University Hospital, 9000 Ghent, Belgium; lieve.brochez@ugent.be

**Keywords:** COVID-19, melanoma, Breslow, incidence

## Abstract

(1) Background: COVID-19 had a major impact on cancer diagnostics and treatment. Delays in diagnosis of cutaneous melanoma were particularly feared, given the impact on survival and morbidity that comes with advanced stages. Moreover, its incidence in Belgium has been rapidly increasing in recent decades. This Belgian population-level study quantifies the pandemic effect on the number of melanoma diagnoses and Breslow thickness in 2020 and 2021. (2) Methods: In using an automated algorithm, the number of cutaneous melanoma diagnoses and Breslow thickness were extracted from all pathology protocols from 2017–2021 by the Belgian Cancer Registry. Monthly variations, as well as year-to-year differences, were studied. (3) Results: Annual incidence of cutaneous melanoma fell by 1% in 2020, compared to 2019, mainly due to a diagnostic deficit in March, April, and May 2020. An 8% incidence increase occurred in 2021, primarily reflecting an increase in the number of the thinnest melanomas (≤1 mm). Both the mean and median Breslow thicknesses were higher in spring 2020, resulting from an underrepresentation of thinner tumors. However, no particulars stood out on a full-year basis in either 2020 or 2021. (4) Conclusions: Considering the expected incidence increase, we estimate almost 210 melanoma diagnoses were missed in Belgium in 2020, corresponding to 6% of the expected number. This deficit occurred mainly during the first COVID-19 wave. Despite some rebound, the 2021 total was still 3% short of the expected number, leaving around 325 diagnoses remaining pending in 2020 and 2021, corresponding to a two-year deficit of 4.35%. Fortunately, mainly thin melanomas were missed, without any detectable shift toward thicker tumors later in 2020 and or 2021.

## 1. Introduction

Already in the first weeks of the COVID-19 pandemic, the number of cancer diagnoses dropped drastically worldwide [1,2,3,4]. Regarding skin cancer, delay of cutaneous melanoma (CM) diagnosis and treatment was particularly feared given the potential impact on survival as well as the economic burden of treating thicker or advanced stages [5,6]. Indeed, CM is only the third most common form of skin cancer but by far the most deadly, accounting for 80% of all skin cancer deaths [2]. Prognosis correlates strongly with Breslow thickness, with stage I corresponding to invasive melanomas less than 2 mm thick without ulceration or less than 0.8 mm thick with ulceration, stage II including all thicknesses beyond that, and stage III implying metastasis to regional lymph nodes. A recent population-based study confirms that treatment delays of three months or more for stages I, II, and III could lead to significant increases in melanoma-specific mortality, with this association being particularly striking for thin, stage I tumors [7,8].

In addition, the incidence of both invasive and in situ CM has been increasing in Belgium in recent decades to a greater extent than for any other solid tumor, accompanied by a growing annual economic burden [9,10]. Between 2004 and 2019, the average annual CM incidence in Belgium was estimated to be just above 22 per 100,000 person-years. With an average annual increase of about 5%, the absolute number of annual CM diagnoses roughly increased two and a half-fold in this time period [11,12]. As such, CM grew to become the fifth most frequent type of cancer in Belgium, i.e., when excluding nonmelanoma skin cancer [13].

Despite increasingly intensive, mainly non-organized screening, there was a slow mortality increase and parallel incidence increase of thick lesions in the last few decades in Belgium, roughly until 2013 [11,14]. Thereafter, overall mortality decreased and stagnated around three per 100,000 person-years in recent years, again despite the implementation of newer treatments (e.g., immunotherapy, targeted therapy) for advanced stages [6,11,13]. The global COVID-19 pandemic represented a new, major, complicating external factor, which undoubtedly had an unprecedented impact on screening and several other facets of regular oncological healthcare. Whereas almost eighty percent of all CM diagnoses come as stage I tumors with reassuringly good 10-year survival rates of up to 95%, a pandemic-related delay might shift this towards more late-stage diagnoses. For example, significantly more sentinel lymph nodes in two German dermatology clinics were found to be positive in 2020 and 2021, compared with 2019, although this did not translate into significantly different AJCC tumor stages [15].

For now, more than three years after the sudden onset of the COVID-19 pandemic and with a fully COVID-19-adapted healthcare system, we wonder what the pandemic’s impact on the incidence and thickness of CM in Belgium may have been and still is. In this study, we provide a population-level quantification of these epidemiological changes, for which the monthly numbers of CM diagnoses and the evolution of reported Breslow thickness values are two parameters of interest. The Belgian Cancer Registry is considered a minimally biased source to investigate this matter because reporting cancer diagnoses is mandatory in Belgium for the pathology laboratories as well as for the oncology care programs [2]. Moreover, CM is particularly suited for this purpose, as diagnosis and subtyping are made in an easily reproducible and reliable manner, with Breslow thickness being consistently reported in pathology protocols, thus facilitating automatic extraction using text mining techniques. To our knowledge, this is the first Belgian population-level study to investigate the impact of the COVID-19 pandemic on CM incidence and thickness.

## 2. Materials and Methods

In this descriptive, observational population-level cohort study, data extraction was carried out by the Belgian Cancer Registry using all pathology protocols of the past five years (2017–2021), with histological diagnosis 8720-8790/3 and skin topography code C44.x (classified according to the International Classification of Diseases for Oncology (Third Edition; ICD-O-3)) [16].

A rule-based algorithm based on pattern matching was used to automatically extract Breslow thickness from the pathology reports of invasive cutaneous melanomas [17]. The algorithm was developed to produce a single label per melanoma diagnosis. For this study, a new melanoma diagnosis was defined as the thickest valid (i.e., recognized by the algorithm, and thus “non-UNK”, see below) Breslow prediction of patient-specific sets of protocols within a one-year period—starting from the earliest protocol for each patient. To estimate the algorithm’s ability to extract the correct Breslow value for each diagnosis, an independent test set was obtained consisting of all protocols from 100 patients randomly selected from the dataset (2017–2020). Two independent expert annotators provided tumor-level Breslow values in millimeters (assuming millimeters when no units were provided in the protocol). The annotators were asked to provide the largest Breslow thickness for each tumor in case multiple values were reported. The inter-annotator agreement was evaluated at 95%. After obtaining consensus labels for the discordant cases, the test set was used to obtain estimates of the algorithm’s performance by considering it a binary classification problem, with true positive and true negative predictions as the correct Breslow thickness in millimeters (two significant figures) and a correctly predicted “UNK”, respectively (see Appendix A). The algorithm itself used a two-step approach. First, a Breslow value prediction (in millimeters) was made per individual protocol (protocol-level extraction). When no Breslow value was found, the algorithm returned “UNK” (unknown).

In a second step, protocol-level extractions were further aggregated into a single extraction per tumor diagnosis, with the latter defined by the set of protocols in a one-year period, with the tumor-level incidence date and Breslow value being, respectively, becoming the earliest and largest protocol-level value for this tumor.

The total numbers of monthly and annual CM diagnoses were studied. We categorized Breslow thickness (in millimeters) with cut-offs based on the AJCC T-staging 8th revision ([0, 1], ]1, 2], ]2, 4], and ]4, infinite] representing tumor stage T1, T2, T3, and T4, respectively), to allocate the results from different years or months and get a brief overview of differences in tumor thickness [18]. We deliberately omitted the additional, newest TNM-defined 0.8 mm stratification level, as its use would result in too small groups [19]. Monthly median and mean Breslow values were plotted for different years to evaluate the effect of two major lockdowns (April 2020 and November 2020). Using the ICD-O-3 classification, the proportion of certain cutaneous melanoma subtypes, as well as the distribution of Breslow thickness within these subtype groups, were also analyzed for all registration years.

## 3. Results

The algorithm’s performance was estimated by using an independent test set of 100 tumor protocols from 100 randomly selected patients, as compiled by two independent annotators. Accuracy was estimated at 95%, with an estimated specificity, precision and recall of 92%, 97%, and 96%, respectively. Its major flaw was the inability to correctly generate a prediction when “Breslow” was not mentioned in the protocol (which was the case for 3 out of 100 patients within the test set).

A total of 27,842 protocols resulted in 21,211 protocol-level extractions, of which 16,640 non-UNK tumor-level Breslow predictions, i.e., new invasive cutaneous melanoma diagnoses. Excluded protocols (i.e., UNK tumor-level) involved either follow-up protocols (biopsy, re-excision) or protocols related to metastases, among others, where the Breslow value is not applicable. The proportion of unknown Breslow values was comparable for different years (around 0.21) and similar to values found in population-based studies covering the preceding two decades [20].

Table 1 shows the exact annual numbers from different years. From 2017 to 2018 and from 2018 to 2019, the annual number of diagnoses increased by 4.45% and 6.32%, respectively, in line with the rising Belgian incidence trends of recent years [13], whereas a downward trend of −1.00% from 2019 to 2020 can be observed. From 2020 to 2021, a rise of 7.97% was noticeable. Annual mean and median Breslow values appear to be similar among all years.

Figure 1 shows the evolution in both the monthly mean and median Breslow values for different registration years. For clarity, the monthly averages of 2017, 2018, and 2019 are taken together in one curve. The graph showing each year as a separate curve can be found in the Appendix A. The mean and median Breslow values of April 2020 were 2.49 mm and 0.70 mm, as compared to an average of 1.44 mm and 0.75 mm in previous years, respectively. Regarding the other months of 2020 as well as 2021, monthly mean and median Breslow values were not substantially different from previous years.

Figure 2 displays the monthly and annual distribution of Breslow values over the past five years. As compared to the same months in 2019, a decline of around 16% in the number of diagnoses was first noticed in March 2020 (221 vs. 263), reaching a 50% drop in April (116 vs. 230) and the number in May was 14% short of May 2019 (257 vs. 298). Thereafter, the monthly number of diagnoses continued to be normal or higher than in previous years. Note: An alternative representation in which breakdown according to Breslow groups rather than months is used and in which absolute numbers of monthly diagnoses can also be read more accurately can be found in the Appendix A. As the histogram on the right suggests, from 2017 to 2019, the annual number of diagnoses increased: furthermore, with a greater increase in the 2018–2019 transition as compared to 2017–2018 (202 or +6.32% vs. 136 or +4.45%, respectively). This seems to be mainly due to an increase in the thinnest melanomas (Breslow thickness [0, 1]). Regarding the 2019–2020 transition, the total number of annual diagnoses decreased by −1.00%; a marginal increase is seen only for the thinnest subgroup ([0, 1]) (1.01%, as compared to +7.40% and +6.38% in 2017–2018 and 2018–2019, respectively). In 2021, as compared to 2020, an increase in the number of annual diagnoses was seen for all Breslow subgroups, thus leading to an increase of almost +8% in the total number of annual diagnoses. Compared to 2019, 2021 mainly shows a sharp increase of diagnoses within the thinnest subgroup (+10.20%), while the numbers for the other groups remain virtually unchanged (+1.00%, −1.70% and +0.78% for subgroups ]1, 2], ]2, 4] and ]4, infinite], respectively). Note: Absolute annual numbers for different Breslow groups can be found in the Appendix A.

Table 2 shows the number of diagnoses for CM subtypes (data only shown for superficially spreading melanoma, SSMM, and nodular melanoma, NM) and according to mean and median Breslow thickness for registration years 2017–2021, the mean Breslow value of NM seemed to be thicker in 2020 as compared to other years. However, after correcting for an extreme outlier (one NM with a Breslow of 100 mm, diagnosed in January 2020), it turned out to be in line with the mean of other years. There were no differences between the mean Breslow values of SSMM for different registration years. Median Breslow values of both subtypes are comparable among all registration years.

## 4. Discussion

As one of the rather few nationwide population-based studies, this study shows a profound impact of the COVID-19 pandemic on the number of new CM diagnoses in Belgium, with a lower-than-expected annual total by the end of 2020.

Whereas annual increases in newly diagnosed cases of 4.45% and 6.32% were seen from 2017 to 2018 and 2018 to 2019, respectively, the actual number of cases in 2020 decreased by −1.00%, as compared to 2019. Indeed, this contrasts sharply with the annual incidence increase of around 5% that would have been expected based on Belgian cancer incidence data, thus suggesting an underdiagnosis due to the COVID pandemic of around 6% [10,12,13,20]. With this in mind, the expected number of diagnoses for 2020 was projected to be around 3570 cases, so with the current number of 3360, an estimated 210 cases remain undiagnosed. Interestingly, a similar number of missed diagnoses (*n* = 279) was observed in a US study involving a population less than half the size of the Belgian population (4.7 million vs. 11.5 million inhabitants) but with similar pre-pandemic incidence rates, which highlights the complexity of influencing factors [21].

In order to put these results into further perspective, it is important to note that the pandemic impacted practically all types of cancer, as described by Peacock et al. [2]. Although the relative decrease in the number of annual melanoma diagnoses in 2020 compared to 2019 was comparable with that seen in cancers such as prostate and breast cancer, it is nevertheless more alarming because CM has been solid cancer with the strongest rising incidence of around 5% in recent years [20,22,23]. By comparison, the expected 2019–2020 evolution in the incidence of, for example, prostate cancer and breast cancer was relatively milder, with a decrease of 1.7% and ‘only’ an increase of 1.2%, respectively [20].

In addition, this study demonstrated a catch-up in 2021 with an incidence increase of almost 8% compared to 2020. However, this, too, should be framed. After all, from 2017 to 2019, the annual number of diagnoses increased by about 11.05%, whereas from 2019 to 2021, it increased by only 6.89%. In other words, the 2021 annual total is estimated to be more than 3% lower than expected (3630 vs. 3745), leaving a shortfall of around 115 diagnoses in 2021 as well. Thus, despite a reasonably uninterrupted restart of routine medical care in 2021, the observed catch-up seems to be incomplete, with an estimated 325 diagnoses remaining pending in 2020 and 2021 (6990 vs. 7315), corresponding to a two-year deficit of 4.35%.

To understand when patients were overlooked, it is essential to take a critical look at the 2020 timeline of monthly diagnoses. A marked decline started in March, reaching the lowest levels in April, with a 58% drop as compared to April 2019. This low corresponds to the first wave of COVID-19 infections and the accompanying first and most stringent lockdown (from 18 March 2020 until 18 April 2020) [24]. This resulted in 31% fewer diagnoses in the period from March to May 2020 compared to the same period in 2019, a periodic shortfall also reported in smaller studies from France (15% [25]), Italy (35% [26] to 60% [27]), and the Netherlands (30%) [28]. A subsequent recovery then set in, reaching near-normal monthly incidences by the end of May and possibly even slightly overcompensating thereafter, resulting in only 2% fewer diagnoses in the period from March to September 2020 compared to the average of the three previous years (1891 in 2020 vs. 1939 on average in 2017–2019). Although the Euromelanoma Campaign 2020 was canceled, the well-known seasonal variation, with typically an increase of diagnoses during summer, might have accelerated this catch-up [29,30,31,32,33]. This course has already been described in other studies [29,34,35].

Nevertheless, two population-based English studies could not demonstrate such a rapid recovery of monthly diagnoses, with a shortfall of more than 20% remaining by November 2020 [4,36]. The number of diagnoses made is inevitably related to available healthcare services, which in 2020 were significantly driven by the burden of COVID-19 infections and restrictions issued by the authorities. For example, the number of diagnoses in April fell more dramatically in the United States than in Canada or most European countries [37]. Likewise, the second wave of infections (October–November 2020) in Belgium had, as shown in this study, little or no impact on the number of diagnoses nor on the distribution of Breslow values because routine care—given precautionary measures—was still provided. Based on the current data, we can therefore state that, in Belgium, particularly in the period from March until May 2020, diagnoses were missed.

On the other hand, one could ask what impact this perceived pandemic-induced delay in diagnosis had on the Breslow thickness, which, although being the most important prognostic factor in primary melanoma, is often lacking in epidemiological studies [14]. Regarding April 2020, an unusually high mean Breslow thickness of 2.49 mm stands out (as compared to an average of 1.44 mm in 2017–2019), whereas the median of 0.70 mm remained well comparable with previous years (an average of 0.75 mm in 2017–2019). This should be carefully interpreted as the consequence of an underrepresentation of thinner tumors rather than a location shift of the entire distribution towards higher Breslow values. After all, in April, the number of T1 and T2 melanomas fell by −60% and −71%, respectively, compared to decreases of −53% and only −21% for T3 and T4 melanomas, respectively. Interestingly, other monthly as well as annual median and mean values for both 2020 and 2021 remained strikingly similar to previous years, which was also one of the key findings in a Dutch population-based study [38]. In contrast, a smaller study in the New York metropolitan area showed a striking increase in mean Breslow thickness in the three months following the first major lockdown (i.e., June, July, and August 2020), which was, however, due to the mere increase of very thick, outlying CMs (Breslow thickness > 4 mm), leaving the median Breslow value within this period virtually unchanged [34]. This finding, which suggests that such delay may have had an impact on a small subset of thicker tumors, applied more or less to our study as well, with around 80 T4 CMs with a Breslow thickness of 4 mm or more, as compared to an average of 63 in June, July and August of previous three years. However, on average, within this period, CMs were not substantially thicker in 2020 than in previous years (mean Breslow thickness of 1.56 mm, as compared to 1.55 mm for 2017–2019).

To detect a possible effect on the typically faster-growing nodular melanomas, we also examined monthly and annual mean and median values for this CM subtype. After correction for an extreme outlier with a Breslow thickness of 100 mm, diagnosed in January 2020 (i.e., before the emergence of COVID-19 in Europe), we found no substantial differences in Breslow thickness on both a monthly and annual basis. However, these numbers rely on CM subtype coding, which is only available for about 30% of all pathology reports.

Considering 2020 as a whole, there seems to be mainly a deficit of the thinnest group (Breslow subgroup [0, 1]). No conclusions could be drawn for the thicker groups with the available data, given the small numbers and large fluctuations in the preceding years.

The 2021 Breslow data were eagerly awaited to nevertheless detect any shift in the distribution of thicknesses. Although compared to 2020, all 2021 Breslow subgroups did show an expected increased trend, absolute numbers of the thicker subgroups were—fortunately—not greater than in pre-pandemic times. Thus, for the time being, the diagnostic delay caused by the COVID-19 pandemic has not resulted in melanomas with higher Breslow thickness at group-level in these cross-sectional data, which was also the conclusion in the latest Dutch population-based study, as well as in German and Italian clinic-based observational studies [15,28,38,39]. However, whether the delay resulted in higher Breslow thickness for individual patients cannot be confirmed. On the one hand, these findings could indicate that the observation period was not sufficiently long to detect an effect on the average thickness of newly diagnosed lesions. As such, it is possible that thin CM lesions grow too slowly for the effect of the delay to show itself yet. Available data suggest the average Breslow thickness increase be somewhere between 0.20 mm and 0.40 mm per month once CM enters the invasive phase, which is, however, based on retrospective and patient memory-dependent estimates, with the growth rate of the primary tumor also showing large inter-case differences [7,38,40]. Our findings, if applicable, could at least suggest a slower growth of the thinnest lesions, which, in Belgium, make up the vast majority of missed diagnoses and have also been the driver of rising incidence rates in recent years [9,13].

On the other hand, it might also be necessary to account for the COVID-induced excess mortality. That is, a number of patients in whom a diagnosis was missed might have died due to COVID-19 and, thus, will never be diagnosed. The latter has been suggested before, especially given the epidemiological overlap between cancer patients and those at risk for a severe COVID-19 course [2,41]. Melanoma risk increases with age, with about 70% of all melanomas in Belgium being diagnosed in patients aged over 50. Patients in the age range of 50–69 years seem to account for the largest share of the observed incidence increase in recent years, followed by patients of 70 years or higher [13].

There are, however, some limitations to this study. Its descriptive, retrospective nature makes drawing preliminary conclusions more difficult. Indeed, many factors introduce a “noise” that may hide any effects of a diagnostic delay: late diagnoses are distributed across multiple groups, with individual delays and actual growth rates remaining unknown. Therefore, estimating the clinical impact of the underdiagnosis remains difficult. In the best-case scenario, the 210 patients missed in 2020 were diagnosed as early as 2021, with seemingly no impact on Breslow thickness. In the worst-case scenario, these patients still did not get their diagnosis, and an additional 115 patients joined this unlucky pool. Only further observation can provide conclusive information on the effects of such delays. On a methodological level, the applied Breslow extraction strategy results in a single patient-level value, i.e., patients with multiple melanomas were represented only once, with the incidence date not necessarily corresponding to the final incidence date. Also, we deliberately omitted the additional, newest TNM-defined 0.8 mm stratification level, as its use would result in too small groups. Finally, this study reports only incidence and thickness data, so extrapolation to the real patient-relevant impact of the pandemic on morbidity and mortality is not possible, while it is precisely in the field of treatment in advanced stages that progress in research has not stood still [39].

## 5. Conclusions

In summary, our study reaffirms that the number of CM diagnoses in 2020 was lower than expected as a result of COVID-19, stressing the importance of taking a rising incidence in recent years into account. This deficit arose mainly in March and April, paralleling the strict lockdown issued by the authorities, and was followed by an incomplete catch-up in 2021 despite a full relaunch of normal healthcare. This incidence decrease seems to apply mainly to thinner T1 tumors and did not result in an increase in thicker tumors for the time being.

## Figures and Tables

**Figure 1 biomedicines-11-01645-f001:**
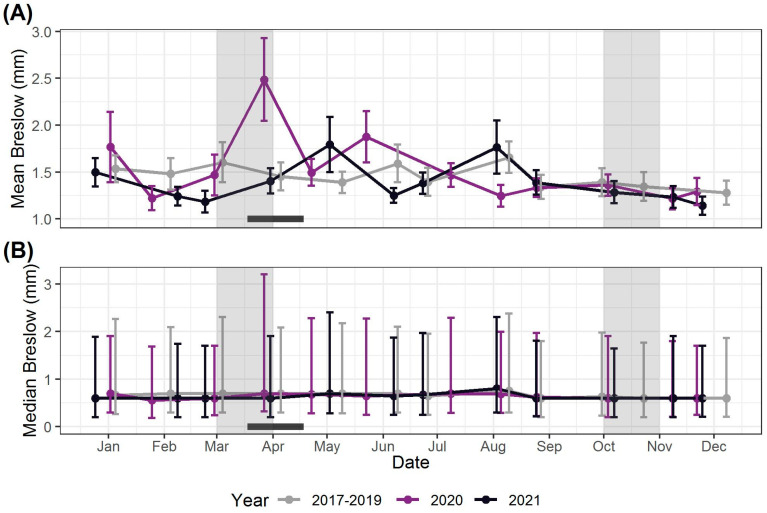
Evolution in monthly mean (**A**) and median (**B**) Breslow value for different registration years (monthly averages of 2017, 2018, and 2019 are taken together in one curve). The 25th and 75th percentiles are shown with brackets. In (**B**), random jitter was added to the *x*-axis values to improve clarity. The grey areas correspond to the periods in 2020 in which an exponential growth in the number of COVID-19 infections was observed; the bold black line corresponds to the period in 2020 in which the stringent lockdown took place.

**Figure 2 biomedicines-11-01645-f002:**
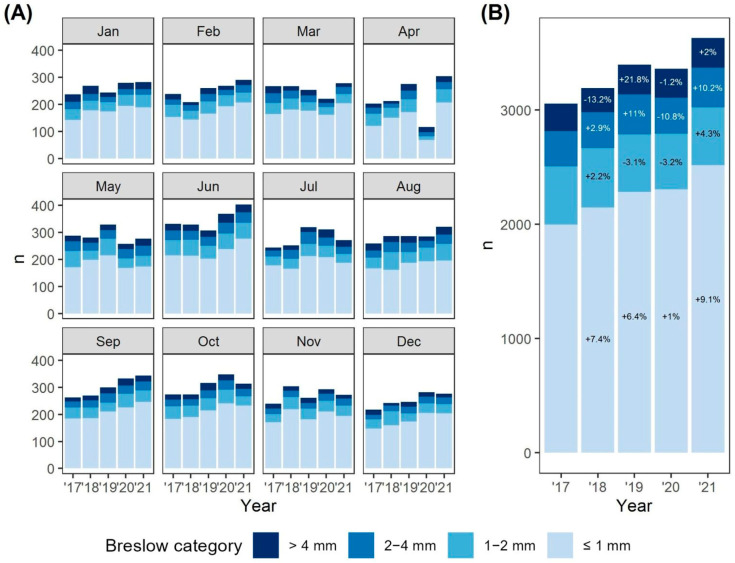
Monthly (**A**) and annual (**B**) distribution of Breslow-values among the melanoma diagnoses over the past five years. (**A**) A decline in de number of diagnoses during the first wave of COVID-19 infections and accompanying stringent lockdown can be seen (March till May 2020). (**B**) Relative differences as compared to the previous year are shown for all Breslow groups separately as well. From 2017 to 2018, an increase in the number of annual diagnoses in the first three Breslow groups ([0, 1], ]1, 2] and ]2, 4]), and a decrease in the thickest subgroup (]4, infinite]) can be observed (total annual increase of +4.35%). From 2018 to 2019, an increase in the number of annual diagnoses in Breslow groups [0, 1], ]2, 4] and ]4, infinite], and a decrease in the subgroup ]1, 2] are seen (total annual increase of +6.32%). From 2019 to 2020 transition, a marginal increase in the number of annual diagnoses is seen for the thinnest subgroup ([0, 1]) only (total annual decrease of −1.00%). In 2021, as compared to 2020, an increase in the number of annual diagnoses is seen for all Breslow subgroups (total annual increase of almost +8%).

**Table 1 biomedicines-11-01645-t001:** Summary of core annual findings.

Years of Registration	2017	2018	2019	2020	2021
Total number of tumor-level Breslow values (i.e., new diagnoses; *n* = 16,640)	3058	3194	3396	3362	3630
Annual percentage increase of tumor-level Breslow values * (%)	NA	4.45	6.32	−1.01	7.97
Mean Breslow value (mm)	1.49	1.45	1.44	1.47	1.38
Median Breslow value (mm)	0.70	0.67	0.68	0.63	0.63

* Percentage change in the number of tumor diagnoses as compared to the previous year.

**Table 2 biomedicines-11-01645-t002:** Melanoma subtype-specific distribution of the number of diagnoses and accompanying mean and median Breslow thickness for different registration years. * After correcting for an extreme outlier (Breslow thickness 100 mm, January 2020, *n* = 1).

	2017	2018	2019	2020	2021
Total number of diagnoses	3058	3194	3396	3362	3630
Total number of diagnoses with a specific histology code	1812	1942	2208	2057	2077
**Nodular melanoma**					
Number (relative to total, percentage)	140 (11.2)	133 (10.6)	123 (10.4)	115 (8.8)	159 (10.2)
Mean Breslow in mm (standard deviation in mm)	4.48 (3.76)	4.29 (3.73)	4.79 (4.71)	5.5 (10.19) 4.73 (5.02) *	4.47 (3.46)
Median Breslow in mm (interquartal range in mm)	3.66 (3.96)	3.32 (3.90)	3.50 (3.47)	3.20 (3.90)	3.65 (4.10)
**Superficial spreading melanoma**					
Number (relative to total, percentage)	889 (71.3)	874 (69.8)	761 (64.1)	846 (64.8)	1059 (68.2)
Mean Breslow in mm (standard deviation in mm)	1.01 (1.40)	0.94 (1.27)	1.06 (1.53)	1.00 (1.34)	0.91 (2.75)
sMedian Breslow in mm (interquartal range in mm)	0.6 (0.63)	0.6 (0.57)	0.6 (0.63)	0.6 (0.60)	0.6 (0.50)

## Data Availability

The data that support the findings of this study are available from the corresponding author upon reasonable request.

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
