# Peer review of "Impact of the COVID-19 Pandemic on the Incidence and Thickness of Cutaneous Melanoma in Belgium"

_biomedicines, 2023, doi:10.3390/biomedicines11061645_

Round 1

Reviewer 1 Report

First of all, I would like to thank you for inviting me to review the manuscript entitled: “Impact of the COVID-19 pandemic on the incidence and thickness of cutaneous melanoma in Belgium”.

The manuscript gives an interesting description of the COVID-19 impact on cutaneous melanoma diagnosis in Belgium. The manuscript is well written in terms of clarity, style, and use of English and has a logical construction. The discussion section explains the findings of the study in the context of published information. The conclusions accurately and clearly explain the main clinical message. The figures are of good quality and relevant to the clinical message. The references are appropriate and current.

I can detect no major flaws in the manuscript.

The only correction I would suggest is that Table 1 should be erased on page 4 since it is also shown on page 3.

The quality of English language is fine.

Author Response

Dear Reviewer,

Yours sincerely,

Philip Georg Demaerel

Reviewer 2 Report

This is a paper on the impact of the COVID-19 pandemic on the incidence and thickness of cutaneous melanoma in Belgium.

It is for the most parts well written. The epidemiological tools and and methodology are appropriate. The data appears to be properly analyzed. The conclusions are supported by the data. 

My only critique is the introduction and discussion are very narrow and dry.

The general reader would like to have more extended overview of the melanoma problem and overview of pathological classification. The latter could include requirements in synoptic reporting. In the discussion I suggest to discuss more in depth limitations.

The above weaknesses are considered as minor and easy to address.

Author Response

(The authors gave the same response as above.)
